# Five Typical Stenches Detection Using an Electronic Nose

**DOI:** 10.3390/s20092514

**Published:** 2020-04-29

**Authors:** Wei Jiang, Daqi Gao

**Affiliations:** School of Information Science and Engineering, East China University of Science and Technology, Shanghai 200030, China; weijiang403880049@163.com

**Keywords:** stenches detection, odor concentration, electronic nose, machine learning algorithm

## Abstract

This paper deals with the classification of stenches, which can stimulate olfactory organs to discomfort people and pollute the environment. In China, the triangle odor bag method, which only depends on the state of the panelist, is widely used in determining odor concentration. In this paper, we propose a stenches detection system composed of an electronic nose and machine learning algorithms to discriminate five typical stenches. These five chemicals producing stenches are 2-phenylethyl alcohol, isovaleric acid, methylcyclopentanone, γ-undecalactone, and 2-methylindole. We will use random forest, support vector machines, backpropagation neural network, principal components analysis (PCA), and linear discriminant analysis (LDA) in this paper. The result shows that LDA (support vector machine (SVM)) has better performance in detecting the stenches considered in this paper.

## 1. Introduction

With the development of China, people pay more attention to environmental problems. The air pollution problem is the most concerning environmental problem in China. Exposure to pollutants such as airborne particulate matter and ozone has been associated with increases in mortality and hospital admissions due to respiratory and cardiovascular disease [1]. So, detecting the low concentration of odors is important. The triangle odor bag method is an olfactory method to measure odor concentration [2]. The specific detection method is recorded in GBT 14675 [3]. Firstly, we need three plastic bags. Two of the three bags are injected with clean air, and the other one is injected with the odor sample. Secondly, the panelist needs to recognize the bag injected with the odor sample. Thirdly, the odor concentration is decreased until the panelist makes a wrong answer. The odor concentration for this panelist is his odor threshold value. Finally, the leader of the panel uses the empirical formula and measured data provided by at least six panelists to determine the odor concentration. But we can see this method has at least three problems: (1) This method needs people to have the ability to smell. (2) Long-term work is harmful to health. (3) The accuracy of the odor concentration depends on the state of the panelists. With the wide application of AI, we decided to use an electronic nose and machine learning algorithms to discriminate stenches.

The term “electronic nose” is often associated with the detection of odors or the attempt to “smell” with a technical device [4]. Today the electronic nose is always used in the evaluation of the food quality [5,6,7,8,9], the evaluation of the quality of fruit [10,11], the classification of wine [12,13,14,15] and other areas. As for feature extraction, different people have different ideas. Zonta et al. [16] use the maximum value of the derivative (Dmax) of the response curve as one of the features of the odor data. Jia et al. [17] use the average value of the data acquired from 60 to 150 s as the features of the odor. We see many papers use the value of the response curve when the derivative of the response curve is about zero, much like the odor feature. So, these important features are also used in our paper.

This paper introduces an electronic nose to classify five kinds of stenches, which is mentioned in GBT 14675. These five chemicals producing stenches are 2-phenylethyl alcohol, isovaleric acid, methylcyclopentanone, γ-undecalactone, and 2-methylindole. The concentration of the chemical solution is based on GBT 14675. This paper will show the algorithm analysis and specific experiments in detail. At the end of this paper, we will compare machine learning algorithms with each other. These algorithms are random forest, support vector machine, PCA (support vector machine (SVM)), LDA (support vector machine (SVM)), LDA (random forest (RF)) and the backpropagation neural network [18,19,20]. The result shows LDA (support vector machine (SVM)) has better performance to detect stenches.

The remainder of this paper is structured as follows: Section 2 describes the materials, the electronic nose, and experimental procedures. Section 3 shows the comparison of machine learning algorithms. Section 4 presents the concluding remarks.

## 2. Materials and Methods

### 2.1. Materials

In GBT 14675, it mentions the concentrations of five kinds of the standard solutions. Table 1 demonstrates the standard concentration recorded in the GBT 14675 and experimental concentration limited to experimental conditions. In order to get the solution, we separately put 3 uL 2-phenylethyl alcohol, 0.3 uL isovaleric acid, 0.95 uL methylcyclopentanone, 0.95 uL γ-undecalactone, and 6 mg 2-methylindole into 30 mL of pure water.Figure 1 shows the actual chemicals and sampling tools.

Because of the difficulty of weighing 0.3 mg 2-methylindole, the concentration of 2-methylindole is higher than the standard concentration. We use 6 mg 2-methylindole to ensure that the measured weight is precise, and the concentration of 2-methylindole is the minimum concentration in the experimental environment.

### 2.2. The Device and Experiment

Figure 2 exhibits the constructed electronic nose [21,22], which consisted of a test box, a clean air cylinder, a personal computer (PC), and two thermostatic headspace vapor generators [23]. The electronic nose uses 9 TGS sensors, specifically TGS 2600, 800, 2602, 2603, 822, 823, 2611, 826, and 832, to detect the odors. When odors interact with the sensing layers of TGS sensors, the sensors will change the resistance value and the voltage value showed in the screen will vary immediately. Different odors will make different sensors output different voltage values. So, we can use the different voltage values as features to classify the odors.

Compared to the old electronic nose mentioned above, we change the sensors types in sensor array to detect the odors. As for these five stenches, we find the voltages change little in negative pressure. So, we use two needles. One needle is to sample the odors, the other is to input the air. We adjust the experimental process and acquisition mode. We use the new algorithm and new hardware to solve the detection problem. Table 2 demonstrates the sensors chosen to detect odors.

Firstly, we set the temperatures of the thermostatic chest and thermostatic cup to 38 degrees centigrade. We set the sampling time to 0.1 s and the measure period is set to 120 s. Secondly, we put the beaker with the solution into the thermostatic cup after the real temperature of the thermostatic cup reaches 38 degrees centigrade. Thirdly, the lift device will move up to let the needle pierce silica gel to sample the odor. Fourthly, a micro vacuum pump extracts gas at 500 mL/s. Finally, we get a 1200 × 9 array to represent the odor. During the experiment, we has a consistent temperature. Sampling with automatic equipment keeps the measure period consistent.

### 2.3. The Composition of Dataset and Feature Extraction

At first, all data subtract the value of the first data to reduce the influence of the experimental environment. Then, we calculate the integral of each curve. The integral of each curve can describe the shape of the curve. Next, we need to use the max value of the curve as the feature. After that, we get the maximum difference of data separated by 20 s as the feature, which is similar to the derivative (Dmax) of the response curve. So, we get 27 features in this way. Figure 3a exhibits a sample of 2-phenylethyl alcohol. Figure 3b exhibits a sample of γ-undecalactone. From these two pictures, we can see that the sensors (826, 2602, 2603) will contribute to the classification of 2-phenylethyl alcohol and γ-undecalactone.

### 2.4. Pattern Recognition Methods

#### 2.4.1. Random Forest

Random forest is an algorithm using an ensemble learning method for classification. Random forest uses random features and random samples to construct the decision tree. Different trees classify the samples using random features, and the features to build the tree will be reformed by training. The sample is classified by the decision trees using the principle of the minority being subordinate to the majority. Random forest has many advantages in classification. It can run on large datasets and high-dimensional datasets. It always has good accuracy in classification and improves the selection of the features.

#### 2.4.2. Back-Propagation Neural Network (BPNN)

BPNN is a widely used tool to classify the samples. BPNN is supervised learning and we can easily use it to deal with the problem of multi-class classification. BPNN consists of forward propagation and back propagation. In the process of forward propagation, we construct a neural network and use the activation function to get a prediction label. The differences between the prediction label and the real label are used to modify the weights of the neural network. BPNN always has good accuracy in classification, but it usually takes a large amount of time to train the neural network. If we do not have a large number of data, the neural network will not be trained well and will have bad accuracy.

#### 2.4.3. Support Vector Machines (SVMs)

SVM is proposed by Cortes and Vapnik, which is widely used in classification and regression problems [24]. SVMs are supervised learning using some generalized linear classifiers to classify the samples. The decision boundary is the maximum margin hyperplane for learning samples. The standard form of SVMs has two types: hard margin and soft margin. However, SVMs have many improved algorithms to adapt to different environments, and we use OVR SVMs or OVO SVMs to deal with the problem of multi-class classification.

#### 2.4.4. Principal Components Analysis (PCA) and Linear Discriminant Analysis (LDA)

PCA is widely used in dimensionality reduction. Firstly, we calculate the covariance matrix of samples. Then, we calculate the eigenvector of the covariance matrix. Finally, we project data into the space formed by eigenvectors. We can clearly discover the distribution of the data using PCA. We also can use classifiers to classify the samples in the eigenvector space. When we reduce the dimension of data to two dimensions, we can use the plot function in Matlab to visual the sample distribution. LDA uses a scatter matrix between classes and a scatter matrix within classes to calculate the eigenvector. We use the eigenvector to reduce the data dimension. Then, we can use simple classifiers to classify the samples. 

### 2.5. Data Initialization and K-Fold Cross-Validation

The randperm function in Matlab initializes data. We use the bsxfun function in Matlab to normalize the feature. We use K-fold cross-validation method to calculate the training set accuracy and the testing set accuracy. We reorder the data by using the random function. K-fold cross-validation reduces variance by averaging the results of K different training groups to lessen the influence of the data partition. The K is 5 in this paper because of the small size of the dataset.

## 3. Results and Discussion

### 3.1. Analysis of Dimension Reduction

In this section, we use PCA and LDA to reduce the dimension of the data. We try to find the distribution of the data and observe whether the samples can be accurately classified in 2D. During the experiment, we find that some sensors will have a big change in voltage value than other sensors. We also discover that some sensors will change a lot in different parts of the process of the sampling stage. Figure 4a shows that all sensors will have a considerable variation in voltage value in 1–60 s. Figure 4b shows that two sensors (2602, 826) will have greatly changed in voltage value in 1–120 s. We need to note that all sensors detecting methylcyclopentanone have large voltages.

During the experiment, we find that sensors (2602, 826) are sensitive to 2-methylindole and methylcyclopentanone. Sensor 826 is also sensitive to γ-undecalactone and 2-phenylethyl alcohol. Figure 5 uses the no. 3 sensor and no. 8 sensor to draw the response curve in 1–120 s. We must note that the blue full line, green full line, and blue full line coincide with the red full line. Because the shape of the curves is similar, we find that only using these two sensors will be difficult to classify isovaleric acid and steam, but these two sensors will contribute to the classification of 2-methylindole and methylcyclopentanone.

During the experiment, we find some odors will cause a huge response from the sensor in different parts of the measure period. Specifically, some of the reaction curves of γ-undecalactone go up quickly in 40–120 s. However, most of the reaction curves of methylcyclopentanone change little in 40–120 s. So, we divide the measure period into two parts. One part is from 1 to 60 s and the other part is from 60 to 120 s.

Figure 6a indicates the distribution of the samples that are sampled in 1–60 s. The first two components, PC1 and PC2, account for 93.32% of the data variance. Figure 6b indicates the distribution of the samples that are sampled in 60–120 s. The first two components, PC1 and PC2, account for 91.08% of the data variance. The first two components of PCA are all bigger than 85%. So, reducing the data dimension to two dimensions remains the feature information. We also find that 2-methylindole, methylcyclopentanone can be easily classified using simple machine learning algorithms. The γ-undecalactone is mixed with 2-phenylethyl alcohol and isovaleric acid.

Then, we use the grid search method to find the best penalty parameter and kernel parameter gamma for the SVM using RBF kernel. The specific multi-classification tool is SVMs using libsvm function in Matlab. We use 5-fold cross-validation ten times to get the training set accuracy and testing set accuracy. The standard deviation of the classification rate is the average of 10 times of 5-fold cross-validation.

For the dimensionality reduction data sampling from 1 to 60 s, the best penalty parameter is 4 and the kernel parameter gamma is 32. The training set accuracy is 91.93%, the standard deviation of the classification rate is 1.95%. The testing set accuracy is 75.70%, the standard deviation of the classification rate is 7.91%. For the dimensionality reduction data sampling from 60 to 120 s, the best penalty parameter is 2 and the kernel parameter gamma is 64. The training set accuracy is 92.75%, and the standard deviation of the classification rate is 1.92%. The testing set accuracy is 72.40%, and the standard deviation of the classification rate is 10.51%.

When we use LDA to reduce the data dimension, the eigenvalues of the samples, which are sampled from 1 to 60 s, are(0.9979, 0.9674, 0.9158, 0.8644, 0.3126). When we use LDA to reduce data dimension, the eigenvalues of the samples, which are sampled from 60 to 120 s, are (0.9926, 0.9760, 0.9104, 0.8596, 0.3754). So, we reduce the dimension of data to four dimensions. We use 5-fold cross-validation ten times to get the training set accuracy and testing set accuracy. For the dimensionality reduction data sampling from 1 to 60 s, the best penalty parameter is 2, and the kernel parameter gamma is 2. The training set accuracy is 98.93%, the standard deviation of the classification rate is 0.57%. The testing set accuracy is 95.30%, the standard deviation of the classification rate is 5.18%. For the dimensionality reduction data sampling from 60 to 120 s, the best penalty parameter is 16 and the kernel parameter gamma is 2. The training set accuracy is 99.88%, and the standard deviation of the classification rate is 0.28%. The testing set accuracy is 97.70%, and the standard deviation of the classification rate is 3.14%.

We use SVM to classify the raw data, which does not reduce the dimension of the data. For the dimensionality reduction data sampling from 1 to 60 s, the best penalty parameter is 4 and the kernel parameter gamma is 2. The training set accuracy is 100%, and the standard deviation of the classification rate is 0%. The testing set accuracy is 88.50%, and the standard deviation of the classification rate is 6.88%. For the dimensionality reduction data sampling from 60 to 120 s, the best penalty parameter is 2 and the kernel parameter gamma is 2. The training set accuracy is 100%, and the standard deviation of the classification rate is 0%. The testing set accuracy is 87%, and the standard deviation of the classification rate is 8.03%.

So, we find that using LDA to reduce the dimension of data is more helpful when classifying the stenches than using PCA. A measure period between 60 and 120 s performs better when using SVM and LDA. Moreover, only using SVM will have a good training set accuracy and a bad testing set accuracy. PCA does not improve the performance of SVM in the experiments. So, the combination of SVM and LDA using the samples from 60 to 120 s has good training set accuracy and testing set accuracy. The standard deviation of the classification rate is also the smallest in these three algorithms.

According to the results of the experiments, we find that the samples of steam are also mixed with the samples of isovaleric acid. Figure 7a exhibits a sample of steam. Figure 7b exhibits a sample of isovaleric acid. The reason for the mixture of steam and isovaleric acid is that the concentration of isovaleric acid is too low. The separation of steam and isovaleric acid is very important. It will prove that the isovaleric acid can be detected by our electronic nose. When we use a higher concentration of isovaleric acid, the voltage value will be more different from the steam.

Table 3 displays the training set accuracy and testing set accuracy of the different algorithms used in this dataset. The STD means the standard deviation of the classification rate of the algorithm.

### 3.2. Analysis of the Other Four Algorithms

For these four algorithms, the algorithm flow is similar. First, we use optimization functions to choose parameters. After that, we use 5-fold cross-validation to get the training set accuracy and testing set accuracy. Finally, we provide standard deviation and comparative analysis. The data set consists of 15 samples of 2-phenylethyl alcohol, isovaleric acid, methylcyclopentanone, 2-methylindole, steam, and 25 samples of γ-undecalactone. The representation of each sample is a 1 × 28 matrix, and the twenty-eighth column is the stenches label. So, the data set is a 100 × 28 matrix.

Table 2 proves that BPNN has a bad performance. The learning rate is set to 0.01, the maximum iteration epoch is set to 500, the goal is set to 10^−3^, and the layer is set to 10. We convert the labels to binary. For the data sampling from 1 to 60 s, the training set accuracy is 62.32%, the standard deviation of the classification rate is 13.67%. The testing set accuracy is 51.10%, the standard deviation of the classification rate is 13.26%. For the data sampling from 60 to 120 s, the training set accuracy is 92.55% and the standard deviation of the classification rate is 9.54%. The testing set accuracy is 87.10%, and the standard deviation of the classification rate is 9.58%. The reason for the poor performance is that the size of the dataset is too small. We also find that the training set accuracy and the testing set accuracy are deeply dependent on the separation of the samples.

Random forest uses the optimization function to choose the number of grown trees. We finally use 50 trees to construct the random forest and it performs well. Figure 8 displays the optimization of the number of grown trees. For the data sampling from 1 to 60 s, the training set accuracy is 100% and the standard deviation of the classification rate is 0%. The testing set accuracy is 90.50%, and the standard deviation of the classification rate is 5.34%. For the data sampling from 60 to 120 s, the training set accuracy is 100% and the standard deviation of the classification rate is 0%. The testing set accuracy is 90.30%, and the standard deviation of the classification rate is 8.20%. 

The accuracy of SVM and RF is similar. So, we use the combination of LDA and RF to classify the stenches. For the data sampling from 1 to 60 s, the training set accuracy is 100% and the standard deviation of the classification rate is 0%. For the data sampling from 1 to 60 s, the testing set accuracy is 95.40% and the standard deviation of the classification rate is 4.12%. For the data sampling from 60 to 120 s, the training set accuracy is 100% and the standard deviation of the classification rate is 0%. For the data sampling from 60 to 120 s, the testing set accuracy is 95.60% and the standard deviation of the classification rate is 4.98%.

We find that the accuracy rates of data sampling from 60 to 120 s using BPNN, LDA (SVM), and PCA (SVM) are higher. LDA (SVM) performs better than PCA (SVM) and SVM. It proves that LDA contributes to the classification of odors. BPNN does not perform well in the experiment, but BPNN also proves that different parts of the measure period will have different accuracy rates.

## 4. Conclusions

In this paper, we use an electronic nose to sample five fixed concentration of stenches and use typical machine learning algorithms to classify the stenches. This paper proposes a stenches detection system composed of an electronic nose and machine learning algorithms to discriminate five typical stenches. Firstly, we observe the curve of the stench and pick up the sensors whose voltage changes a lot during sampling the data. We can also separate the measure period into several parts to find the difference between the stenches. Then, we can use PCA and LDA to reduce the dimension of the data to observe the major problem of classification. Finally, we use random forest, support vector machine, PCA (SVM), LDA (SVM), and backpropagation neural network to solve the problem.

We can see this model will be sensitive to the dataset. We find that sensors (2602, 826) are sensitive to 2-methylindole and sensor 826 is sensitive to γ-undecalactone, but only use these two sensors can not classify steam and isovaleric acid. LDA (SVM) improves the accuracy of SVM only using raw data. Random forest and SVM perform well in this dataset. BPNN takes a considerable amount of time to train the net and has poor performance. The accuracy of BPNN depends on the partition of the dataset and the size of the dataset. BPNN may work well when increasing the size of the dataset. So, we suggest using LDA (SVM) to detect these five odors. Compared to the measure period of 1–60 s, the measure period of 60–120 s is easier to classify 2-phenylethyl alcohol, 2-methylindole, γ-undecalactone, methylcyclopentanone, and steam. We can see that this model can replace the parts of panelists’ smelling works. Future research will focus on more kinds of low concentration national standard stenches.

## Figures and Tables

**Figure 1 sensors-20-02514-f001:**
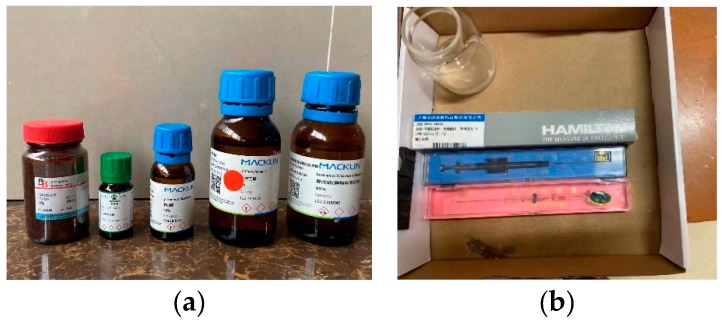
(**a**) Actual chemicals. (**b**) Sampling tools: 1-uL, 10-uL, and 0.5-uL needles and a 250-mL beaker.

**Figure 2 sensors-20-02514-f002:**
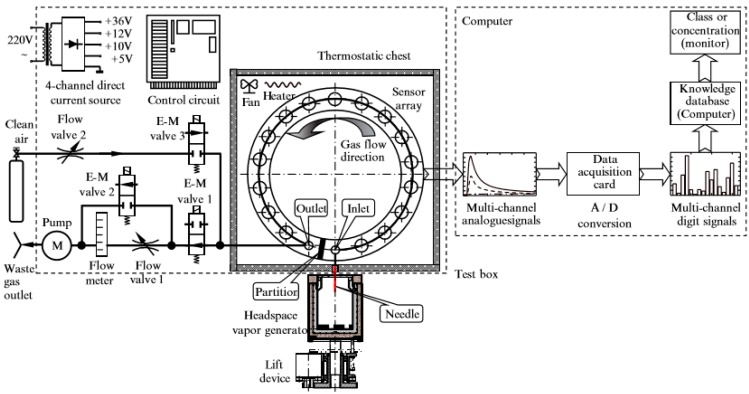
Electronic nose.

**Figure 3 sensors-20-02514-f003:**
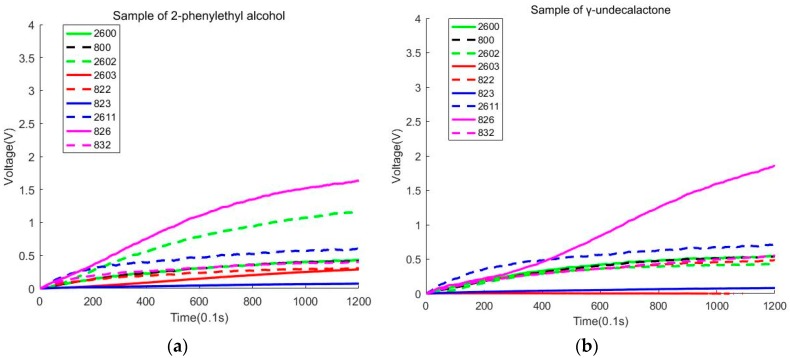
(**a**) Sample of 2-phenylethyl alcohol. (**b**) Sample of γ-undecalactone.

**Figure 4 sensors-20-02514-f004:**
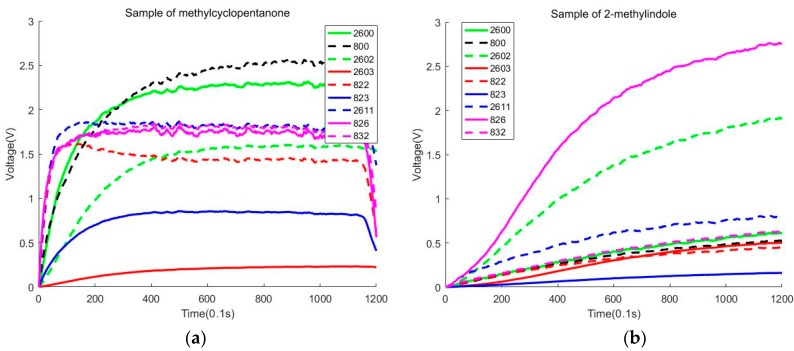
(**a**) Sample of methylcyclopentanone. (**b**) Sample of 2-methylindole.

**Figure 5 sensors-20-02514-f005:**
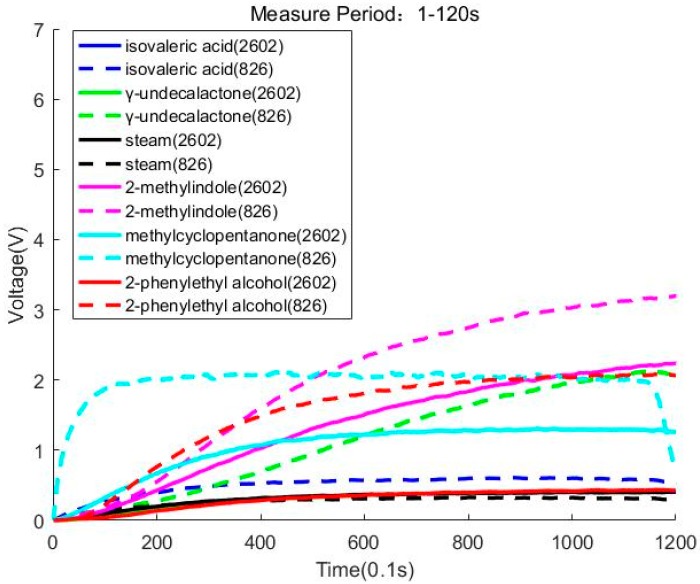
Sampling picture of sensors (2602, 826).

**Figure 6 sensors-20-02514-f006:**
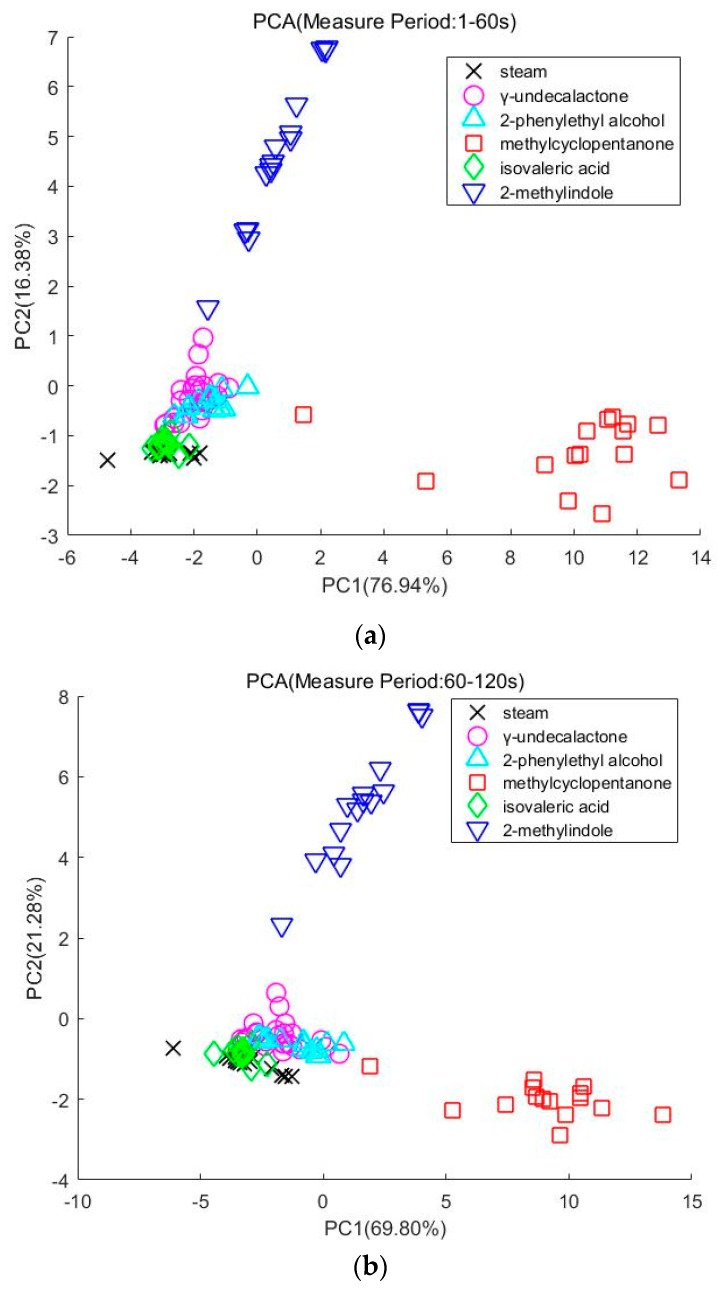
(**a**) PCA (Measure Period: 1–60 s). (**b**) PCA (Measure Period: 60–120 s).

**Figure 7 sensors-20-02514-f007:**
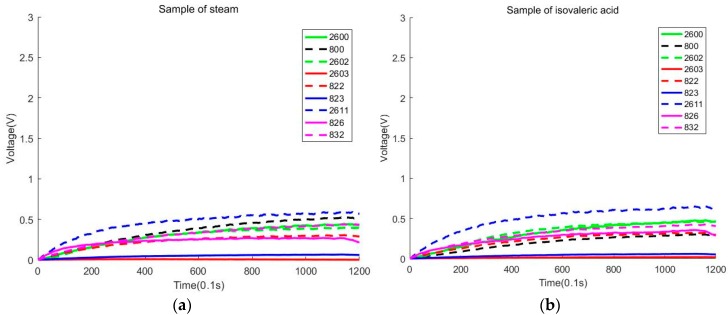
(**a**) Sample of steam. (**b**) Sample of isovaleric acid.

**Figure 8 sensors-20-02514-f008:**
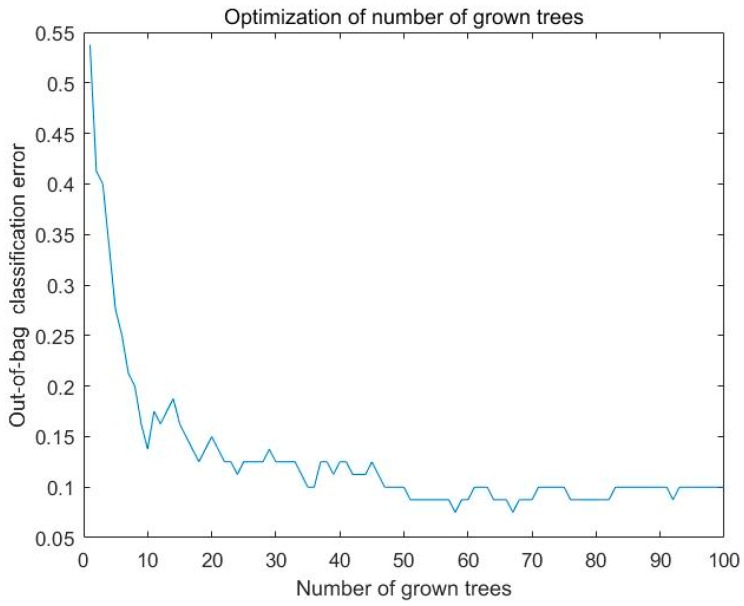
Optimization of the number of grown trees.

**Table 1 sensors-20-02514-t001:** Concentration of five kinds of standard solutions.

Chemical solution	Standard Concentration	Experimental Concentration
2-phenylethyl alcohol	$10^{-4}$	$10^{-4}$
isovaleric acid	$10^{-5}$	$10^{-5}$
methylcyclopentanone	$10^{-4.5}$	$10^{-4.5}$
γ-undecalactone	$10^{-4.5}$	$10^{-4.5}$
2-methylindole	$10^{-5}\left(w/w\right)$	$10^{-4}\left(w/w\right)$

**Table 2 sensors-20-02514-t002:** Type of sensors.

No. in Array	Sensor Name	Typical Target
1	2600	$H_{2}$, CO, etc.
2	800	Combustible gas, etc.
3	2602	Alcohol, methylbenzene, etc.
4	2603	Methyl mercaptan, etc.
5	822	Benzene, etc.
6	823	Isobutane, etc.
7	2611	${\mathrm{CH}}_{4}$, Combustible gas, etc.
8	826	${\mathrm{NH}}_{4}$, etc.
9	832	R22, R134a, etc.

**Table 3 sensors-20-02514-t003:** Comparison of different algorithms.

Algorithm	1–60 s Training Set Accuracy (STD)	1–60 s Testing Set Accuracy (STD)	60–120 s Training Set Accuracy (STD)	60–120 s Testing Set Accuracy (STD)
BPNN	62.32% (13.67%)	51.10% (13.26%)	92.55% (9.54%)	87.10% (9.58%)
RF	100% (0%)	90.50% (5.34%)	100% (0%)	90.30% (8.20%)
LDA (RF)	100% (0%)	95.40% (4.12%)	100% (0%)	95.60% (4.98%)
SVMs	100% (0%)	88.50% (6.88%)	100% (0%)	87.00% (8.03%)
LDA (SVM)	98.93% (0.57%)	95.30% (5.18%)	99.88% (0.28%)	97.70% (3.14%)
PCA (SVM)	91.93% (1.95%)	75.70% (7.91%)	92.75% (1.92%)	72.40% (10.51%)

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
