# Peer review of "Five Typical Stenches Detection Using an Electronic Nose"

_sensors, 2020, doi:10.3390/s20092514_

Round 1

Reviewer 1 Report

The new version of the paper partly answered my previous concerns. However, the presentation still needs more precision to highlight the technical value of the paper.

1)      The first version of the paper states that “We can see the PCA and LDA 14 analysis proposed in this paper have better performance to detect stenches”, while the second version the authors state that “We can see random forest in this paper has better performance to detect stenches”. How can this be explained?

2)      Page 3, line 82. The authors state “we change the sensors types in sensor array to solve the new problem”. Which exactly is the new problem? Which sensors were changed in the sensor array and why?

3)      The graphics in figure 3 are too small. It is impossible to distinguish between colors and shapes of curves. The authors could separate each graphic into several ones with a small number of curves each one. Plotting 16 curves on a small graphic is not useful for interpretation.

4)      The graphics in figure 4 are too small to analyze. The same remark as before.

5)      Page 5, line 151. I do not agree with the remark “Figure 4a displays that all sensors will have a big change in voltage value in 1-60”. It seems that three of four sensors have a small change in voltage value. However, doe to the size of the graphic, which is too small, it is not possible to say which ones.

6)      Page 5, line 152. The authors state that “Figure 4b displays that four sensors will have a big change in voltage value at all times. So we can separate the data into two parts and recognize the sample one by one.” Why one has to separate data in two parts if four sensors have big change … Please explain this statement.

7)      The comments of the authors related to figure 4 b, state that only 4 sensors are selected. Could it be more useful to build a sensor network using only the 4 sensors identified as more performant? Does the other sensors have a significate impact on the performance of the classification?

8)      Page 5, sub-section 2.4. Why does the authors need to randomize the data before performing the 5-fold cross validation?

9)      Figure 5 is too small. It is absolutely impossible to read the legend.

10)   Page 6, line 160. “Figure 5 uses no.6 sensor and no.13 sensor to draw the response curve in 60—120s.” However, the graphic in figure 5 draws much more than two curves.

11)   Figure 6 is too small too.

12)   It is not clear how does the authors use the % variation explained by the first two components.  

13)   Page 6, line 181. The authors state that “Figure 4a shows the reason for the separation of methylcyclopentanone.” The authors must provide clear justifications for their statements.

14)    Concerning the use of the SVM to define the boundary in figure 6d, the authors provide the values of the parameters. However, the authors must provide an accurate description of the data used for training the SVM. If I understood well, they use a 5-fold cross validation approach. For instance, which is the size of the data set?

15)   The graphics in figure 7 are too small. Please consider previous remarks.

16)   Page 8, lines 209, 210. The authors must recall the name of three algorithms they analyze in subsection 3.2.

17)   Page 8, line 226. “The training set accuracy is 0.9904, the testing set accuracy is 0.9913.” It is strange that the test accuracy is better that the training accuracy. Is it correct?

The authors need to improve the precision of the presentation and resubmit the paper. 

Reviewer 2 Report

In “Five typical stenches detection using electronic nose” authors investigate the possibility to correctly identify five different “stenches”, namely vapors which produce a discomfortable sensation in people. The main goal of the manuscript is to investigated the capability of a gas sensor array to replace panelists for detecting the presence of bad smells in the environment. The five chemicals utilized for the scope are 2-phenylethyl alcohol, isovaleric acid, methylcyclopentanone, γ-undecalactone, 2-methylindole. Authors utilized an array composed by 16 commercial sensors and extracted 3 different feature to the signal coming from each of them. Finally, different classification algorithms (such as random forest, support vector machines, backpropagation neural network, PCA and LDA) are used to evaluate the performance of the proposed array. Despite the aim of paper is surely of a great interest for both the scientific community and the health care, in the reviewer opinion the manuscript is not very clear in many parts and some data are lacking. The missing parts make difficult to completely understand the impact of the work. Reviewer suggests to deeply revise the manuscript considering the following observations/comments:

  1. Concerning the compounds measured:
  • Reviewer thinks that the correct name for one of the compound is γ-undecalactone rather than γ-unsecalactone
  • 2-methylindole melting point is 57-59 °C from literature so it seems to be solid rather than viscous ( 1- methylindole is liquid at room temperature).
  • Usually it is convenient to specify the concentration of vapors of samples in terms of part per million (ppm). In this context it is not clear how the sample vapor concentration in the head space is produced (is the liquid heated? is it just let evaporating?). Furthermore, even if the concentration in liquid is known, it is not reported which is the concentration of analyte vapour in the headspace.
  • In case of 2-methylindole, the concentration utilized is higher than the standard one. Why the authors do not perform a further dilution in water in order to obtain the desired concentration?
  • From what reviewer understands, each sample is measured multiple times. Did the authors measure all the repetition of the same sample consecutively or did they randomly measured the sample alternating the different compounds?

Please note that vapour concentration is crucial information in case of gas sensors. Most of the compounds investigated have a very low saturated vapour pressure and concentrations in water are indeed very low.

  1. Concerning the aim of the manuscript, actually the system was tested considering only one concentration for each compound. Despite this approach is utilized to standardize the odour recognition, in a real scenario, analytes are usually present in a range of concentrations rather than at a fixed concentration. This condition makes the detection and recognition of odours even more challenging since the increasing of concentration can produce the overlapping of classes during the classification. In the reviewer opinion the effect of concentrations must be considered in this kind of applications to provide robust outcomes and performances.

  1. Concerning data analysis and classifications, please specify which features have been used in the cases described.
    - Authors utilized 5-fold validation algorithm. How many times authors ran this algorithm? Since the composition of training and test dataset change each time a large number of iteration should be performed. Furthermore mean and standard deviation of classification rates obtained during the different runs of cross-validation algorithm should be both reported.
    - Figure 9 reports an Out of bag classification error equals to 0.15. How do authors can justify this value with the accuracies reported in the manuscript?
    - LDA percentage for LD1 and LD2 should be reported (figure 7 a) and b)) in other to understand the significativeness of results.

  1. Authors utilized as features the responses in the 60-120s period. In this particular condition, authors considered the response value at t=60s as zero. Reviewer wouldlike to make some considerations about this strategy:
    - First of all subtracting value very close each other increases enormously the noise/signal noise. Than this approach is not recommended when the variation of response is close to zero since data are largely affected by noise and lose of significativeness.
    - Secondly the transient responses depends on the concentration profile of analyte and flow. When features not related only to the steady state are used, authors must verify and specify that the flow rate was kept constant during the whole experiment and that the concentration of analytes in the measuring chamber is constant too for the whole measuring period (120 s). Otherwise systematic errors can influence or bias the results obtained.
    Sensors in the array have high responses but the main discrimination power seems to be enclosed in the very small variations obtained close to the steady state. In the reviewer opinion this condition undermines the robustness of results obtained; then this part should be taken higher in consideration and investigated more in details.

Reviewer suggests to deeply check the manuscript, to make it more clear and to provide detailed information about experimental set-up. Furthermore, different concentrations should be investigated if the aim of the manuscript is to develop a system able to replace panelist.

Round 2

Reviewer 1 Report

Globally, the authors provide proper answers to the questions that I raised. However, I recommend them to read carefully their contribution as some typos are still present. See for instance:

1)      Abstract: “Stenches refer to some kinds of gas which can stimulate olfactory organs to discomfort people and pollute the environment in this paper”.  This sentence could be reformulated as “This paper deals with the quantification of stenches, which can stimulate olfactory organs to discomfort people and pollute the environment”.

2)      Avoid starting a sentence with “And …”

3)      Abstract: the sentences “We can see LDA (SVM) in this paper has the better performance to detect 14 stenches. And LDA contributes well to the solution of stenches detection.” Could be replaced by “The results show that LDA (SVM) has better performance in detecting the stenches considered in this paper.”

4)      Minor English typos are still present and need a careful read when preparing the final version of the paper.

I recommend the paper to be accepted for publication.

Reviewer 2 Report

Reviewer appreciates the effort of authors in amending the manuscript. Authors provided most of the missing information in the manuscript even if there are few points that are still not completely clear to the reviewer. 

  • In paragraph "3.2. Analysis of the other Four Algorithms" authors wrote "The data set is consisted of 15 samples of 2-phenylethyl alcohol, isovaleric acid, methylcyclopentanone, 2- methylindole and 25 samples of γ-undecalactone. The representation of each sample is a 1x28 matrix and the twenty-eighth column shows the stench label. So the data set is a 100x28 matrix." if reviewer calculations are correct when samples are summed up the total results to be 85 (15*4+25). Secondly in the text number of feature results to be 27. Please check this part.
  • Concentration of 2-methylindole is actually 2*10-4 (w/w). Please specify in the table that concentrations are reported in w/w ratio.
  • It would be helpful if table 3 could report the standard deviations. Sometimes sigma values are included next to percentage between brackets (e.g. LDA(SVM)  98.93% (0.57%)  95.30% (5.18%)  99.8% (0.28%)  97.70% (3.14%) ). In the reviewer opinion this would be helpful in understanding the statistical relevance of obtained results.
  • As general recommendation, reviewer discourage from measuring consecutively the same sample. It is recommended to alternate the order of sample in order to avoid that drift or systematical errors can influence the results. Random sequence of samples, or random blocks sequence, is always recommended even if this procedure is much more time consuming.  
  • Finally reviewer suggests to carefully read the text trying to correct typos or to remove repetitions in the manuscript. For example:
    - In lines 146-150, "big change in voltage value" is widely repeated
    - line 72 "When odors touch the TGS sensors" - actually odours may interact with the sensing layers of TGS sensors rather than touch it.
    - line 86 - sampling time to 120s. Reviewer understands that sampling time can be misleading but actually in electronics it is referred to the time between the acquisition of two consecutive readings of sensor values. In the case of authors, reviewer thinks that sampling time is 0.1s since in 120s the system acquired 1200 samples. The measure period is indeed 120s.

Author Response

This manuscript is a resubmission of an earlier submission. The following is a list of the peer review reports and author responses from that submission.

Round 1

Reviewer 1 Report

The authors propose an electronic nose based method to detect the odor of five stenches. The problem is well formulated and the research objectives are clear.

However, the presentation of the results must be improved. The authors must provide precisions on the following aspects:

The choice of the sensors must be justified. Why using 16 sensors and why those ones? Figures 4, and 5 are not clear as it is not indicated which response belongs to which sensor. The comments of the authors related to figure 5 b, state that only 4 sensors are selected. Then why considering the other 8? Could it be more useful to build a sensor network using only the 4 sensors identified as more performant for the application addressed in this paper? The same 4 sensors are more performant for all stenches? Regarding the PCA method, the authors should prove that using only 2 principal components provides a good approximation. They can use, for instance, the Cumulative Percent Variance. The performance of the artificial intelligence methods is highly dependent on the parameter setting. The authors must provide the parameters they used for these methods when comparing their performance to the combination of PCA and LDA. For instance, the hyperparameters setting for the SVM method is very important for the performance of the classification. Page 7, line 181: It is not clear why separating data in four parts using PCA and why separating the dataset in three parts using LDA. In figures 6, 7 and 8, the authors should use different symbols for each stench. It is easier to analyze the graphics. 

The paper has scientific value, but it needs an important improvement in the precision regarding both the presentation of the results and the methods used.

Reviewer 2 Report

The submitted paper report the detection and recognition of typical stenches using a MOS-based electronic nose. The reviewer thinks that a key point for the electronic nose is to improve the performance of the MOS sensors instead of to torture the data, as we know “if you torture the data long enough, it will confess to anything”. The reviewer does not see any improvement made by the electronic nose system compared with that reported by the authors 8 years ago [21,22]. Moreover, for such kind of simple odor sensing, the discussion of algorithm more than PCA is generally meaningless if the device itself has poor performance. Therefore, the reviewer cannot recommend this paper for publication in Sensors.